# Prevalence of mortality among mechanically ventilated patients in the intensive care units of Ethiopian hospitals and the associated factors: A systematic review and meta-analysis

**Temesgen Ayenew**[1]*, **Mihretie Gedfew**[1], **Mamaru Getie Fetene**[2], **Belayneh Shetie Workneh**[3], **Animut Takele Telayneh**[4], **Afework Edmealem**[1], **Bekele Getenet Tiruneh**[5], **Guadie Tewabe Yinges**[6], **Addisu Getie**[1], **Mengistu Abebe Meselu**[1]

1 Department of Nursing, College of Health Sciences, Debre Markos University, Debre Markos, Ethiopia,
2 Department of Midwifery, College of Health Sciences, Debre Markos University, Debre Markos, Ethiopia,
3 Department of Emergency Medicine and Critical Care Nursing, University of Gondar, Gondar, Ethiopia,
4 Department of Public Health, College of Health Sciences, Debre Markos University, Debre Markos, Ethiopia, 5 Department of Internal Medicine, School of Medicine, Debre Markos University, Debre Markos, Ethiopia, 6 Debre Markos Comprehensive Specialized Hospital, Debre Markos, Ethiopia

* teme31722@gmail.com, temesgen_ayenew@dmu.edu.et

## Abstract

### Background

In the intensive care unit (ICU), mechanical ventilation (MV) is a typical way of respiratory support. The severity of the illness raises the likelihood of death in patients who require MV. Several studies have been done in Ethiopia; however, the mortality rate differs among them. The objective of this systematic review and meta-analysis is to provide a pooled prevalence of mortality and associated factors among ICU-admitted patients receiving MV in Ethiopian hospitals.

### Methods

We used the Preferred Reporting Items for Systematic Reviews and Meta-Analyses (PRISMA) 2020 criteria to conduct a comprehensive systematic review and meta-analysis in this study. We searched PubMed/Medline, SCOPUS, Embase, Hinari, and Web of Science and found 22 articles that met our inclusion criteria. We used a random-effects model. To identify heterogeneity within the included studies, meta-regression and subgroup analysis were used. We employed Egger's regression test and funnel plots for assessing publication bias. STATA version 17.0 software was used for all statistical analyses.

### Results

In this systematic review and meta-analysis, the pooled prevalence of mortality among 7507 ICU-admitted patients from 22 articles, who received MV was estimated to be 54.74% [95% CI = 47.93, 61.55]. In the subgroup analysis by region, the Southern Nations, Nationalities,

**Data Availability Statement:** All relevant data are within the manuscript and its Supporting Information files.

**Funding:** The author(s) received no specific funding for this work.

**Competing interests:** The authors have declared that no competing interests exist.

and Peoples (SNNP) subgroup (64.28%, 95% CI = 51.19, 77.37) had the highest prevalence. Patients with COVID-19 have the highest mortality rate (75.80%, 95% CI = 51.10, 100.00). Sepsis (OR = 6.85, 95%CI = 3.24, 14.46), Glasgow Coma Scale (GCS) score<8 (OR = 6.58, 95%CI = 1.96, 22.11), admission with medical cases (OR = 4.12, 95%CI = 2.00, 8.48), Multi Organ Dysfunction Syndrome (MODS) (OR = 2.70, 95%CI = 4.11, 12.62), and vasopressor treatment (OR = 19.06, 95%CI = 9.34, 38.88) were all statistically associated with mortality.

## Conclusion

Our review found that the pooled prevalence of mortality among mechanically ventilated ICU-admitted patients in Ethiopia was considerably high compared to similar studies in the United States (US), China, and other countries. Sepsis, GCS<8, medical cases, MODS, and use of vasopressors were statistically associated with mortality. Clinicians should exercise caution while mechanically ventilating ICU-admitted patients with these factors. However, it should be noted that the exact cause and effect relationship could not be established with this meta-analysis, as the available evidence is not sufficient. Thus, more studies using prospective methods will be required.

## Introduction

In the intensive care unit (ICU), mechanical ventilation (MV) is a typical way of respiratory support. Mechanical ventilation is necessary for 20%-40% of admissions to the ICU in the United States, according to the Society of Critical Care Medicine [1]. According to the American Association for the Surgery of Trauma, more than half of ICU patients are ventilated within the first 24 hours of admission [2]. According to a study by A. Anzueto and A. Esteban, 39% to 49% of ICU patients undergo mechanical ventilation at any given moment [3]. The global burden of patients who require mechanical ventilation has increased due to the emerging severe acute respiratory syndrome coronavirus 2 (SARS-CoV-2) viral pandemic [4].

Between 3% and 20% of patients admitted to the hospital with coronavirus disease (COVID-19) require ICU treatment, and many of these patients require MV [5, 6]. Advances in critical care medicine, as well as the use of invasive mechanical ventilation in the ICU, have resulted in better short-term survival and mortality outcomes in critically ill patients [7, 8]. However, the degree of illness of patients admitted to the ICU increases the likelihood of death for patients requiring MV [9]. According to a large cohort study conducted in southern Brazil, the mortality rate of patients who required MV was approximately 51% [10]. Other studies in United States (US), China and an international study reported mortality rates of 29.75%, 35.36% and 28%, respectively [7, 9, 11].

In Ethiopia, the mortality rate of ICU patients receiving MV ranges from 28.6% in Tigray region's Ayder Comprehensive Specialized Hospital to 88.5% in Addis Abeba's St. Paul's Millennium Medical College (SPMMC) [12, 13]. Different determinant factors have been reported in different studies done in Ethiopia. These include the presence of at least one comorbid illness, the length of stay on MV for more than three days, night time admission to the ICU, Glasgow coma scale (GCS) score ≤8 during admission, sepsis, use of vasopressor therapy, admission with medical cases, and multiple organ dysfunctions (MODS). The mortality rate also varies between different ICU settings, locations, and types of patients who were admitted

to ICU and received MV [12, 14–18]. However, data on the nationally representative pooled mortality rate of ICU patients receiving MV in Ethiopia are scarce.

The objective of this systematic review and meta-analysis is to provide a pooled prevalence of mortality and associated factors among ICU-admitted patients receiving MV in Ethiopian hospitals. The findings of this research will serve as a baseline reference for local and national quality improvement activities aimed at improving the survival of ICU-admitted patients using MV. Furthermore, this review will identify the determinant factors associated with mortality in patients admitted to the ICU receiving MV and will provide information to clinicians and researchers to develop strategies to mitigate the effect of the identified determinant factors associated with mortality.

## Methods

### Study design and search strategy

The objective of this systematic review and meta-analysis was to determine the overall mortality rate among ICU patients in Ethiopia who received MV. For this systematic review and meta-analysis, the Preferred Reporting Items for Systematic Reviews and Meta-Analyses (PRISMA) 2020 checklist was used [19]. We ran a thorough search of many international databases, including PubMed/Medline, SCOPUS, Embase, Hinari, and Web of Science, to find published articles. In addition, we searched Google Scholar for unpublished studies and grey literature. All published and gray literature was retrieved, critically analyzed, and assessed for inclusion in this study until December 30, 2023.

To retrieve the articles, the following search phrases were used with "AND" and "OR" Boolean operators: ('mortality' OR "death' OR 'mortality rate' OR 'rates, mortality' OR 'death rate' OR 'rate, death' OR 'rates, death') AND ('associated factors' OR 'predictors' OR 'determinants') AND ('Patients who received mechanical ventilation' OR 'Patients who received Ventilations, Mechanical' OR 'Patients who received Mechanical Ventilations' OR 'Patients who received Ventilation, Mechanical') AND ('Intensive Care Units' OR 'Intensive Care Unit' OR 'Unit, Intensive Care') AND ('Ethiopia' OR 'Federal Democratic Republic of Ethiopia'). The search approach Co, Co, Pop (Condition, Context, and Population) was applied (**S1 Table**).

### Inclusion and exclusion criteria

This review included observational studies (prospective and retrospective cohort, and cross-sectional) that reported the prevalence of mortality among pediatric and adult ICU-admitted patients who received MV in Ethiopia and were publishe or grey literature sources reported in English and published any time before December 30, 2023. Articles from which the prevalence of mortality could be estimated from the data reported in it were also included. However, articles that did not provide full text access, failed to report the prevalence of mortality in mechanically ventilated patients admitted to the ICU, and if the prevalence of mortality could not be estimated from the data within the article were excluded.

### Outcomes

The primary outcome, represented as percentage and frequency in articles, was the prevalence of mortality among patients admitted to the ICU who received MV in Ethiopia. The secondary outcome was factors affecting the mortality of mechanically ventilated patients in Ethiopia, which were represented in the form of odds ratios and or in cross-tabulation as cell values of number of exposed with the outcome, number of exposed without the outcome, number of nonexposed with the outcome and number of nonexposed without the outcome. As a result,

the secondary outcome was provided as odds ratios estimated by meta-analysis of odds ratios from individual studies that reported the determinant variable or the variable's cell values from cross-tabulated data. The variables used in this meta-analysis to estimate the secondary outcome were those that were deemed statistically significant in the primary studies.

### Data extraction and data quality assessment

The titles and abstracts of all the retrieved studies were independently evaluated by two reviewers (T.A. and M.A.M). Full-text review was open to papers that passed the title and abstract screening. Two authors (M.G and A.T.T) reviewed the full-text of eligible studies. The disagreement was resolved using the inclusion and exclusion criteria, and the final decision was made by the third reviewer (A.E).

Microsoft Excel 2019 was used to extract data that included the author's name, publication year, study region, mortality prevalence, sample size, study design, and population category. Two independent reviewers (M.A.M and T.A) extracted and cross-checked the extracted data for any variations, and any inconsistencies were resolved by re-reading the full text.

The quality of the included studies was assessed using the Joanna Briggs Institute (JBI) quality assessment tool for prevalence studies [20]. The criteria to be met were as follows: 1) sample representativeness, 2) adequate sample size, 3) correct measurements, 4) acceptable statistical analysis procedures, and 5) response rates. Studies that met or above 50% of the assessment criteria were deemed low risk of bias.

### Data analysis

To manage the selection process, Endnote version X8 reference management software was used. For data analysis, STATA version 17 was employed. Egger's test and the funnel plot were used to look at potential publication bias [21, 22]. Article heterogeneity was evaluated with $I^2$ statistics [23]. The random effects model of Der Simonian and Laird was used to assess the pooled prevalence of mortality and associated factors with 95% CI among ICU-admitted patients who underwent MV. To identify the source of heterogeneity, a meta-regression and subgroup analysis based on region, ICU type (adult versus pediatrics), COVID-19 status, study design and sample size was performed. The sensitivity test was performed to determine the impact of individual studies on the pooled estimate.

## Results

### Article selection

A total of 1,775 potentially qualifying studies were discovered using the PRISMA flow diagram. After eliminating duplicates and doing titles review, 26 studies were found. Four studies were excluded after full-text review because their primary outcome was incidence of mortality and they lacked data to estimate the prevalence of mortality. Finally, for the final systematic review and meta-analysis, 22 papers were included (**Fig 1**).

### Study characteristics

This review and meta-analysis included 22 studies with a total of 7507 participants that were conducted in Ethiopia and published in indexed journals or identified in grey literature. Seven of the studies were retrospective cohort, four were prospective cohort, and the remaining eleven were cross-sectional. The studies were done in Addis Ababa [12, 15–18, 24–28], Amhara [29–33], Oromia [34], Southern Nations, Nationalities and Peoples (SNNP) region [14, 35–37], and Tigray [13, 38]. The sample size ranged from 104 to 630 (Table 1).

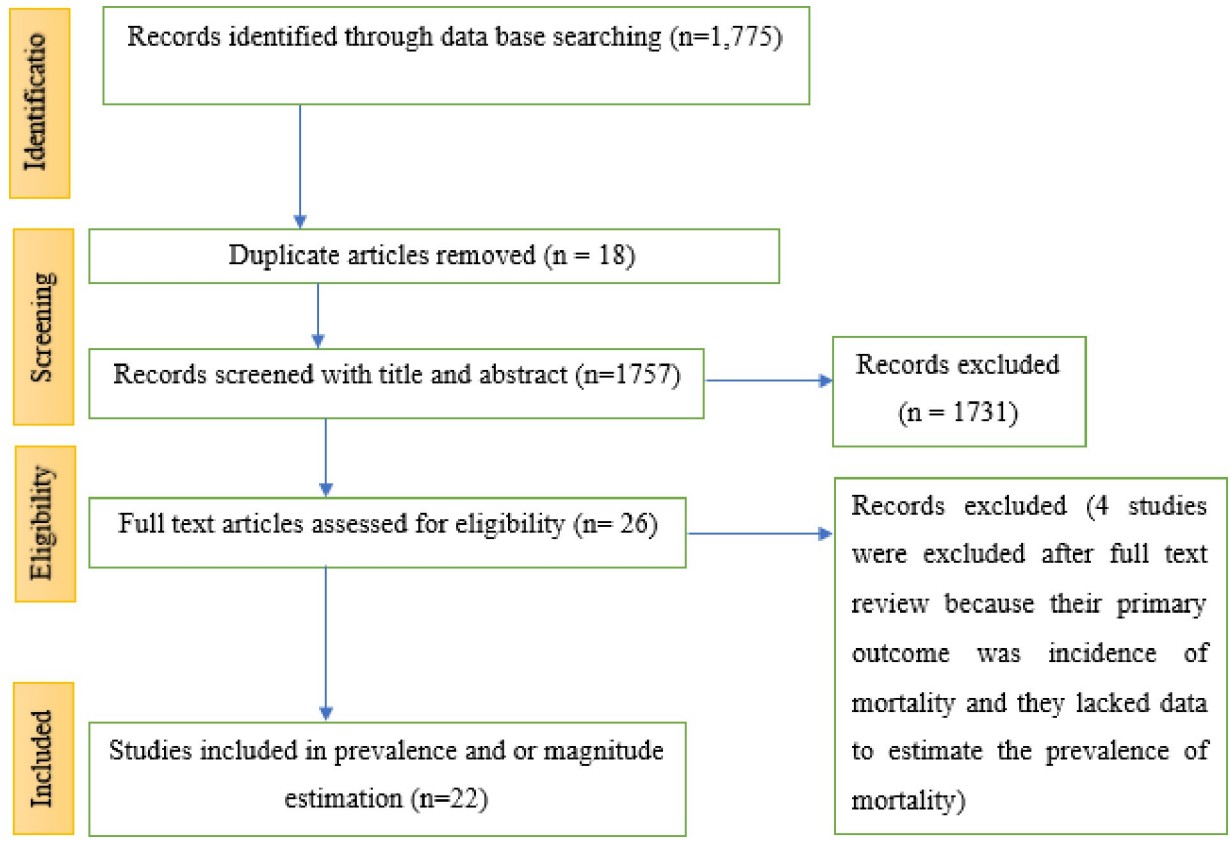

**Fig 1. PRISMA flow-chart depicting the selection process of studies in Ethiopia, 2023.**

**Table 1. General characterisitcs of selected studies for the prevalence of mortality among mechanically ventilated ICU patients in Ethiopia, 2023.**

| Authors name | Region | publication year | Source type | Study design | Type of ICU | Sample size | Prevalence (%) | JBI Quality |
|---|---|---|---|---|---|---|---|---|
| Alemayehu et al. [24] | Addis Ababa | 2022 | Journal | Cross-sectional | Adult | 202 | 41.7 | 9 |
| Debebe et al. [16] | Addis Ababa | 2022 | Journal | Retrospective cohort | Adult | 160 | 60.7 | 9 |
| Berhe, et al. [13] | Tigray | 2023 | Journal | Prospective cohort | Adult | 286 | 28.6 | 9 |
| Tilahun et al. [33] | Amhara | 2022 | Journal | Retrospective cohort | Adult | 376 | 33.78 | 9 |
| Zewudie et al. [32] | Amhara | 2023 | Journal | Retrospective cohort | Adult | 388 | 55.98 | 9 |
| Wotiye et al. [35] | SNNP | 2022 | Journal | Retrospective cohort | Adult | 310 | 80.82 | 7 |
| Bacha et al. [34] | Oromia | 2023 | Journal | Prospective cohort | Pediatrics | 206 | 34.5 | 7 |
| S. Seid et al. [31] | Amhara | 2022 | Journal | Cross-sectional | Adult | 402 | 37.6 | 9 |
| Tigist B. et al. [15] | Addis Ababa | 2021 | Journal | Cross-sectional | Pediatrics | 220 | 59.1 | 9 |
| Hunegnaw et al. [18] | Addis Ababa | 2022 | Journal | Cross-sectional | Adult | 247 | 57.1 | 9 |
| S.M. Abate et al. [14] | SNNP | 2021 | Journal | Retrospective cohort | Adult | 517 | 67.3 | 9 |
| Abate et al. [36] | SNNP | 2023 | Journal | Prospective cohort | Adult | 630 | 49 | 9 |
| Endeshaw et al. [25] | Addis Ababa | 2022 | Journal | Retrospective cohort | Adult | 410 | 77.45 | 9 |
| Seifu et al. [26] | Addis Ababa | 2022 | Journal | Cross-sectional | Pediatrics | 406 | 67.94 | 9 |
| Demass et al. [30] | Amhara | 2023 | Journal | Cross-sectional | Adult | 568 | 51.35 | 9 |
| Dendir et al. [37] | SNNP | 2023 | Journal | Cross-sectional | Pediatrics | 396 | 60.05 | 9 |
| Gemechu E. et al. [28] | Addis Ababa | 2022 | Journal | Cross-sectional | Pediatrics | 260 | 34.5 | 6 |

*(Continued)*

**Table 1.** (Continued)

| Authors name | Region | publication year | Source type | Study design | Type of ICU | Sample size | Prevalence (%) | JBI Quality |
|---|---|---|---|---|---|---|---|---|
| Nega G et al. [27] | Addis Ababa | 2023 | Journal | Retrospective cohort | Adult | 496 | 63.3 | 9 |
| Korbu et al. [12] | Addis Ababa | 2023 | Journal | Cross-sectional | Adult | 104 | 88.5 | 6 |
| Tsegay et al. [17] | Addis Ababa | 2023 | Thesis | Cross-sectional | Adult | 210 | 54.8 | 9 |
| Haftu et al. [38] | Tigray | 2018 | Journal | Cross-sectional | Pediatrics | 400 | 37.5 | 9 |
| Teshager NW, et al. [29] | Amhara | 2020 | Journal | Prospective cohort | Pediatrics | 313 | 62.2 | 9 |

## Prevalence of mortality among intensive care unit patients who received mechanical ventilation

This systematic review and meta-analysis determined that the pooled prevalence of mortality among ICU-admitted patients on MV to be 54.74% [95% CI = 47.93, 61.55]. There was a high inter-study heterogeneity ($I^2$ = 97.5%, P = 0.0001). (**Fig 2**). As a result, a subgroup analysis was done to identify the source of hetrogeneity by study location, sample size category, ICU type, study design and COVID-19 status (COVID-19 versus non-COVID-19 ICUs).

## Subgroup analysis

In the subgroup analysis by region, the SNNP subgroup (64.28%, 95% CI = 51.19, 77.37) had the highest prevalence (**Fig 3**). The subgroup analysis based on sample size shows that studies with sample size >400 results in a higher prevalence [59.15% (95% CI = 49.27, 69.04)] (**Fig 4**). Subgroup analysis by ICU type revealed that the studies conducted in adult ICUs yield the highest prevalence [56.4% (95% CI = 47.87, 65.21)] (**Fig 5**). In subgroup analysis by study design, retrospective cohort studies have the highest prevalence [62.8% (95% CI = 51.15, 74.46)], Whereas, prospective cohort studies show the lowest prevalence [43.63% (95% CI = 29.50, 57.76)] (**Fig 6**). The subgroup analysis of patients with COVID-19 versus non-COVID-19 patients revealed a significant difference in prevalence. COVID-19 patients have the highest mortality rate (75.80%, 95% CI = 51.10, 100.00) (**Fig 7**).

## Sensitivity analysis

A sensitivity analysis was performed to see whether each study had an impact on the pooled prevalence estimates. The results of the sensitivity analysis indicated that, in a random-effects model, no study affected the total pooled prevalence (**Fig 8**).

## Publication bias

According to the results of Egger's test with a p value of (p = 0.5331) and a symmetric funnel plot (**Fig 9**), there was no publication bias.

## Meta-regression

Meta-regression using publication year, JBI Quality and sample size as factors was used to find the cause of heterogeneity. However, the results demonstrated that these variables were not significant enough to be identified as causes of heterogeneity (Table 2).

## Factors associated with mortality

This meta-analysis discovered various determinant factors for mortality in mechanically ventilated ICU-admitted patients in Ethiopia. GCS<8, sepsis, vasopressor use, medical cases, and

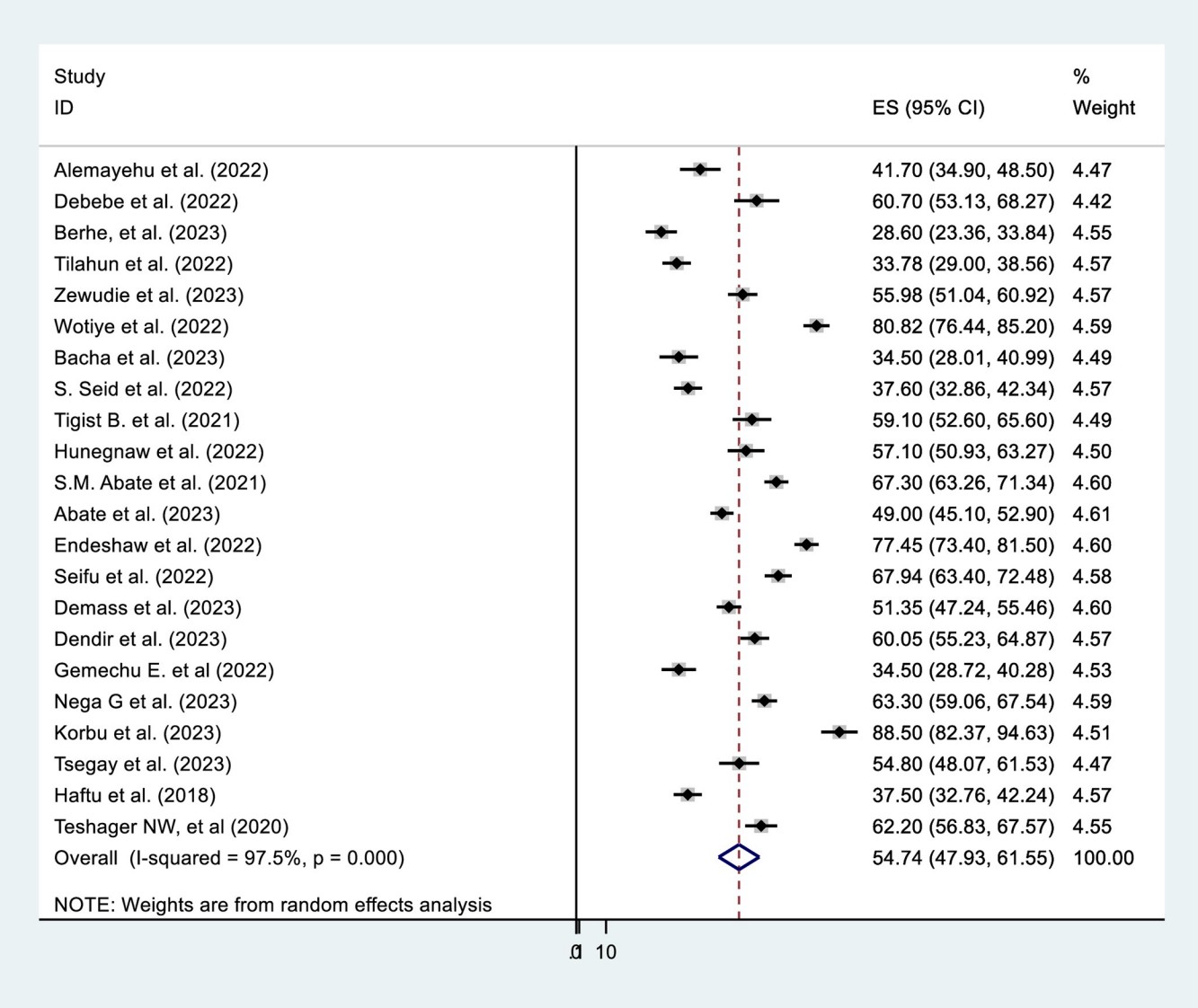

**Fig 2. Prevalence of mortality among intensive care unit patients who received mechanical ventilation in Ethiopia, 2023.**

the presence of MODS were significantly associated with death. Patients on MV who had sepsis were 6.85 (OR = 6.85, 95% CI = 3.24, 14.46) times more likely to die than those who did not have sepsis. Patients who were mechanically ventilated and had GCS<8 at the beginning of ventilation were 6.58 [OR = 6.58, 95%CI = 1.96, 22.11] times more likely to die than those who did not. Mechanically ventilated patients admitted with medical cases were 4.12 [OR = 4.12, 95%CI = 2.00, 8.48] times more likely to die than surgical cases admitted. Patients with MODS and on mechanical ventilation were 2.70 [OR = 2.70, 95%CI = 4.11, 12.62] times more likely to die than those without MODS. Mechanically ventilated patients who received vasopressor medication were 19.06 times more likely to die [OR = 19.06, 95% CI = 9.34, 38.88] than those who did not (Table 3).

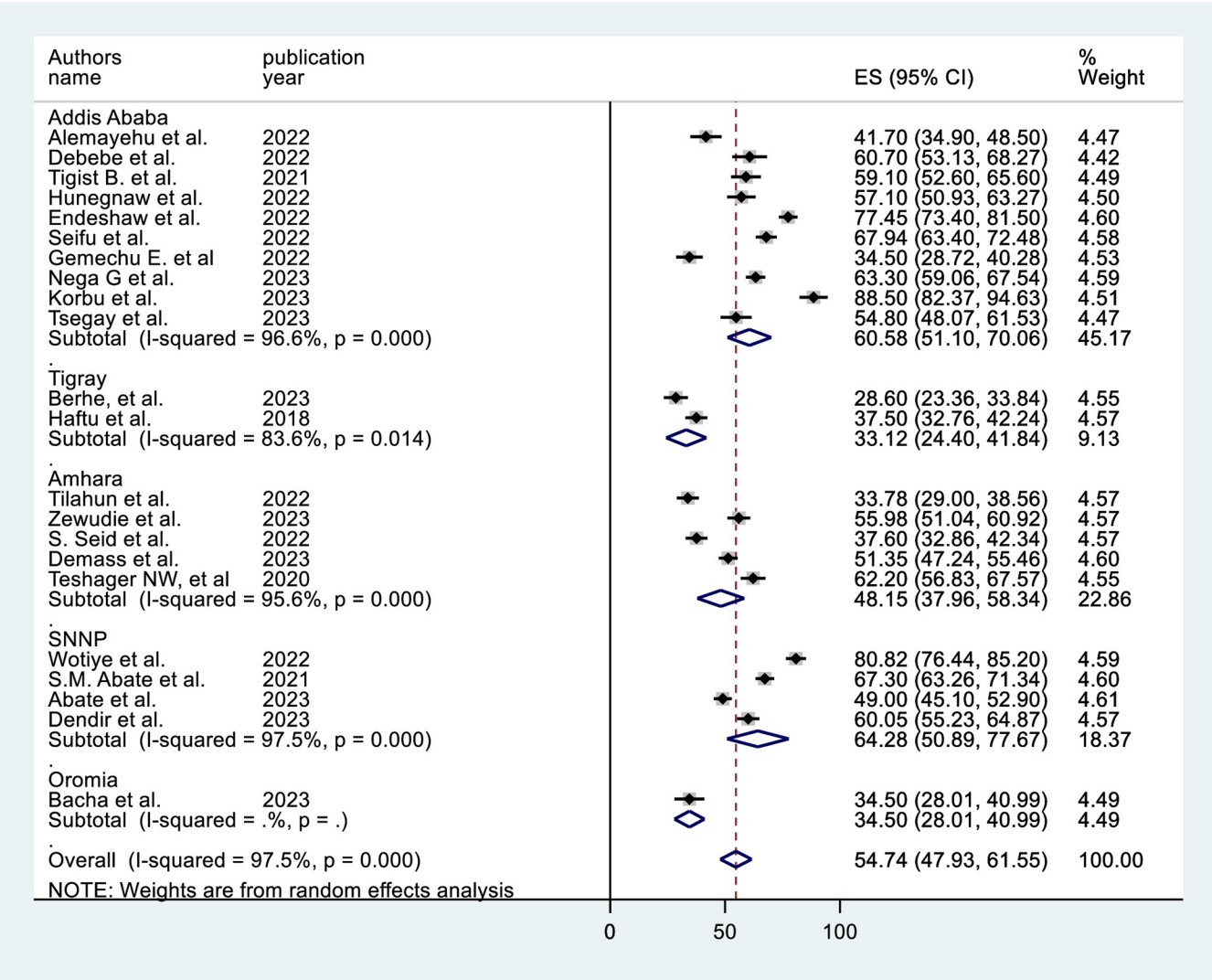

**Fig 3. Subgroup analysis by region of mortality among mechanically ventilated ICU patients in Ethiopia, 2023.**

## Discussion

Maintaining appropriate oxygenation and ventilation while lowering the risk of complications and enabling the patient's lungs to recover or regain function is the major objective of MV. Although MV is not typically thought of as a treatment for acute respiratory failure per se, ventilator management needs to be closely monitored, as improper ventilation could aggravate morbidity and mortality by injuring the lungs or respiratory muscles [39, 40]. The general prevalence of mortality among ICU-admitted patients using MV in Ethiopia was estimated in this systematic review and meta-analysis.

The pooled prevalence of mortality among mechanically ventilated ICU-admitted patients in Ethiopia was determined to be 54.74%, according to our review. Our findings are comparable to those of a Brazilian study (51%) [10]. Our result, however, is higher than those of other similar studies conducted in the US (29.75%) [11], China (35.36%) [9], and an international study(28%) [7]. The disparities could be related to differences in research settings, which could

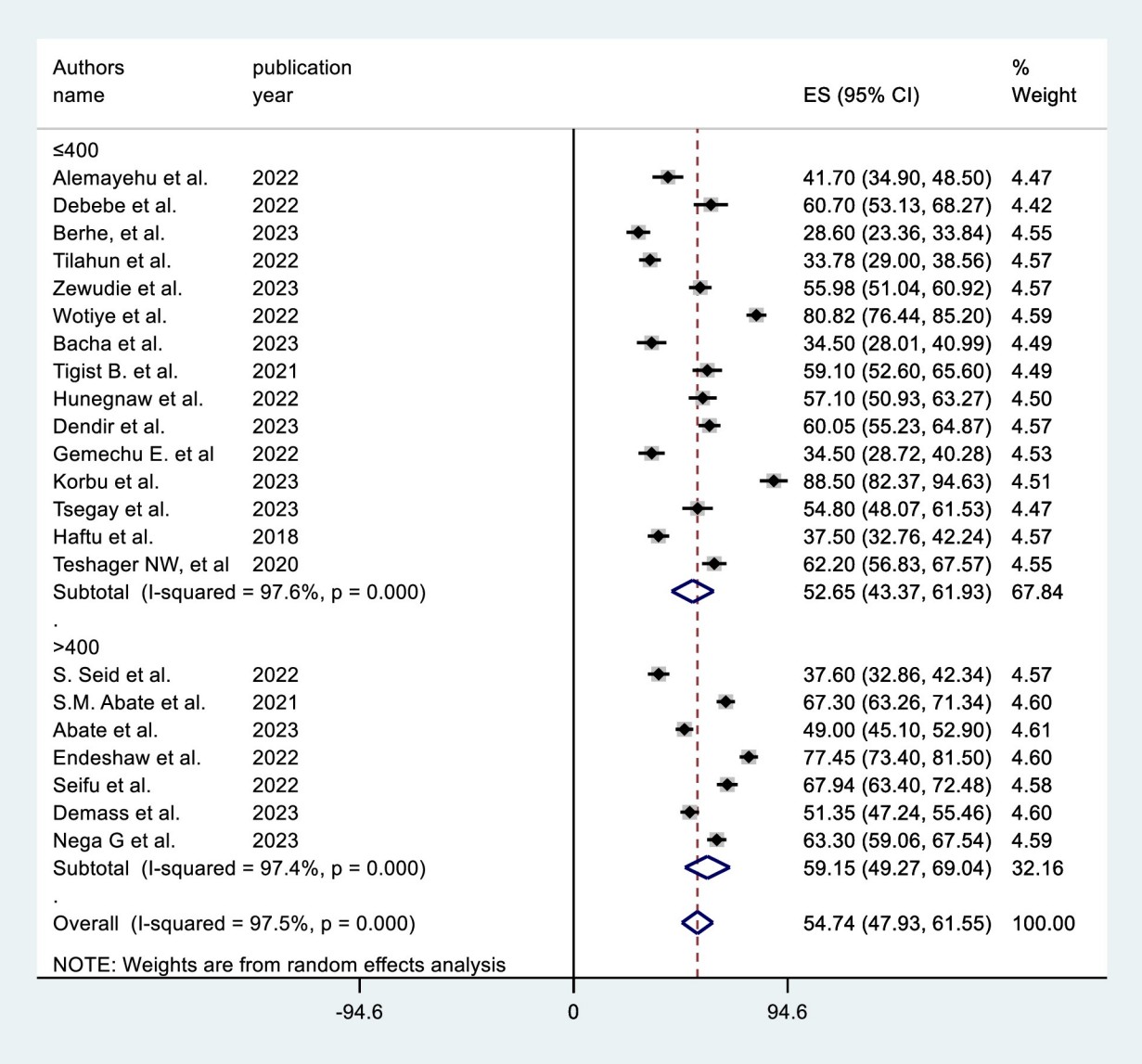

**Fig 4. Subgroup analysis by sample size of mortality among mechanically ventilated ICU patients in Ethiopia, 2023.**

be described in terms of ICU advancements. The ICUs in developed nations such as the United States and China are more modern in terms of equipment and medical personnel than in Ethiopia [41].

In the subgroup analysis, there is a substantial difference in mortality prevalence between the COVID-19 and non-COVID-19 groups. Mortality was observed to be higher in mechanically ventilated COVID-19 patients than in non-COVID-19 patients (75.80% versus 52.64%). A comparable result was observed in another study, with mechanically ventilated COVID-19 patients dying at a rate of 97% [42]. This could be attributed to the acute respiratory distress syndrome and sepsis caused by COVID-19 infection [43, 44].

Compared to the other regions included in the review, the SNNP region had the highest (64.28%) prevalence of mortality. This could be owing to a regional shortage of specialist personnel and other ICU resources. According to a review of critical care services in Ethiopia

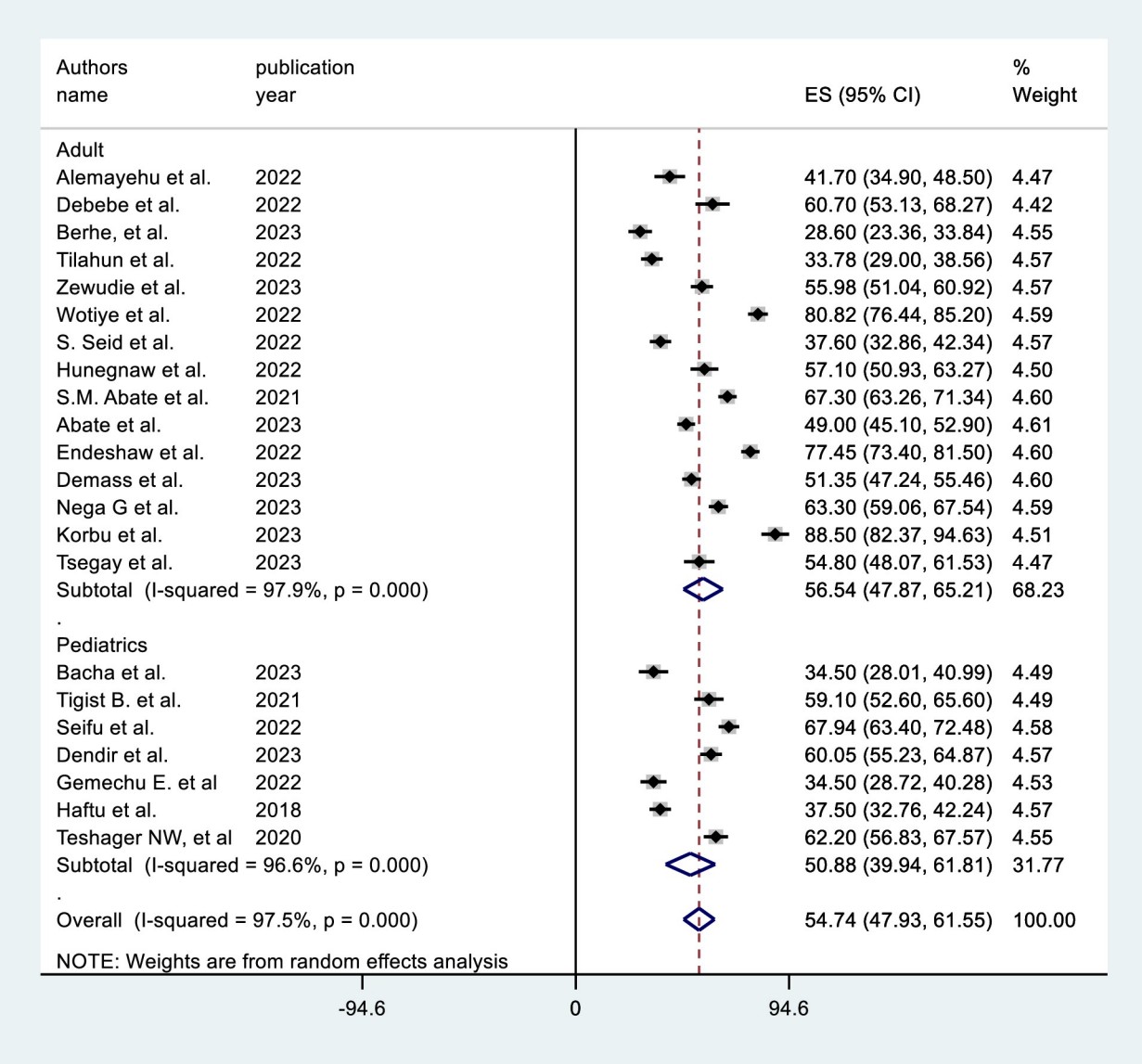

**Fig 5. Subgroup analysis by type of ICU of mortality among mechanically ventilated ICU patients in Ethiopia, 2023.**

released in 2022, not all ICUs in the region had critical care physicians, and the region had insufficient ventilation capacity [45].

According to this meta-analysis, mechanically ventilated individuals with sepsis were 6.85 times more likely to die than those without sepsis. A study by the Japan Sepsis Alliance study group and the American Association for Respiratory Care support this [46, 47]. This could be because MV can induce lung injury, which can aggravate the condition in patients with sepsis. Furthermore, sepsis can result in respiratory failure, which increases the risk of death [48, 49].

Patients on MV with GCS<8 at the start of ventilation were 6.58 times more likely to die than those with GCS>8. Another study found that intubation at admission was linked to increased mortality, ICU days, and overall length of stay in patients with GCS of 6–8 [50]. Tanaka, A. et al., 2022 found that patients with GCS scores < 8 had significantly higher hospital mortality [51]. Unconsciousness during admission could be a sign of a more serious

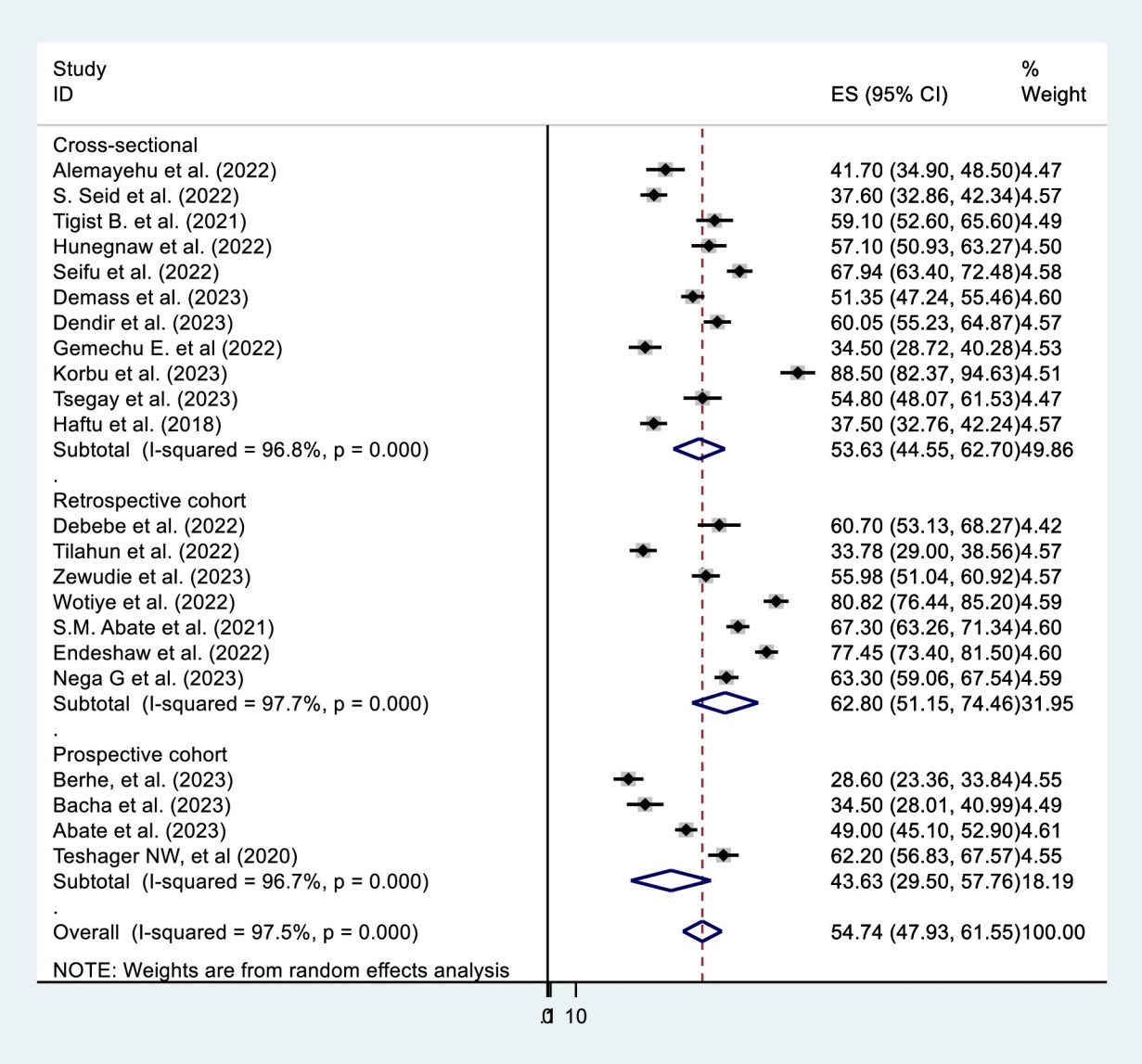

**Fig 6. Subgroup analysis by study design of mortality among mechanically ventilated ICU patients in Ethiopia, 2023.**

condition, leading to a higher mortality rate. However, it is crucial to highlight that the association between unconsciousness and mortality is not always clear and can rely on a variety of factors such as the underlying cause of unconsciousness, the duration of unconsciousness, and the patient's overall health status [52, 53].

The mortality rate for mechanically ventilated patients admitted with medical cases was 4.12 times higher than that of individuals admitted with surgical cases. Other similar studies by Oscar Peñuelas et al. (2021) and the Respiratory Therapy Zone webpage's (2024) report support this [53, 54]. This could be explained by the greater likelihood of chronic respiratory disorders, such as chronic obstructive pulmonary disease (COPD), in those with medical diagnoses [55]. Another explanation is that individuals with medical conditions are often older and have more comorbidities, increasing the risk of complications and death. Patients with

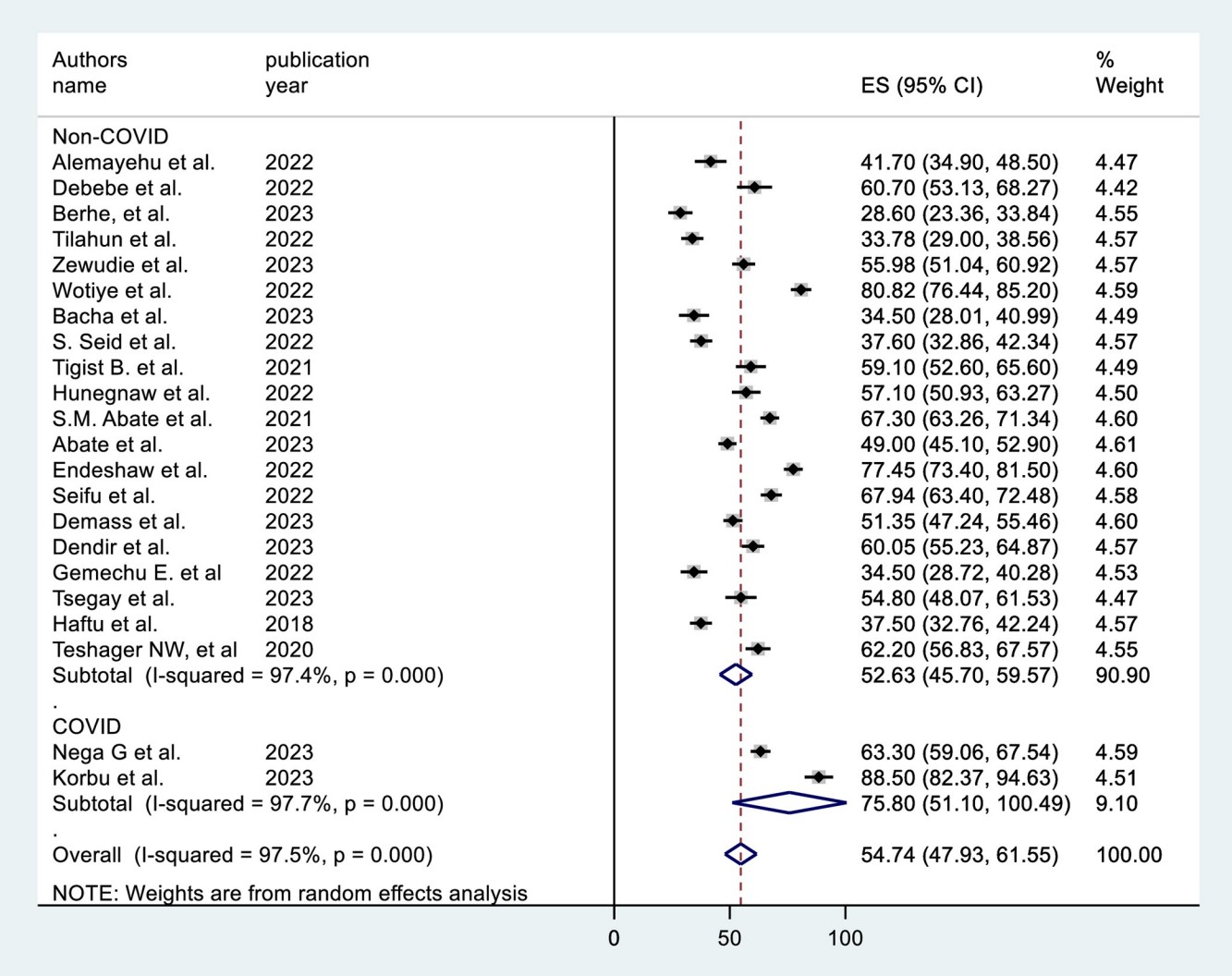

**Fig 7. Subgroup analysis by COVID-19 of mortality among mechanically ventilated ICU patients in Ethiopia, 2023.**

surgical diagnoses, on the other hand, are often younger and healthier, which can contribute to better outcomes [56].

Patients who were mechanically ventilated and had MODS were 2.70 times more likely to die than their counterparts. A comparable report has been found in two different studies: Xiao, K. et al., 2014 and Xiao, K. et al., 2020 [57, 58]. This might be due to the fact that MODS can set off a chain of events that might cause more damage to the body and increase the risk of complications and mortality [58].

Patients on MV who received vasopressor medication were 19.06 times more likely to die than those who did not. Another study conducted in Korea found that the use of vasopressors was strongly related to in-hospital mortality in mechanically ventilated patients [59]. This could be explained by the possibility of ICU-acquired weakness linked to the use of vasoactive drugs [60]. In most cases, ICU-acquired weakness is associated with considerable morbidity and mortality [61, 62].

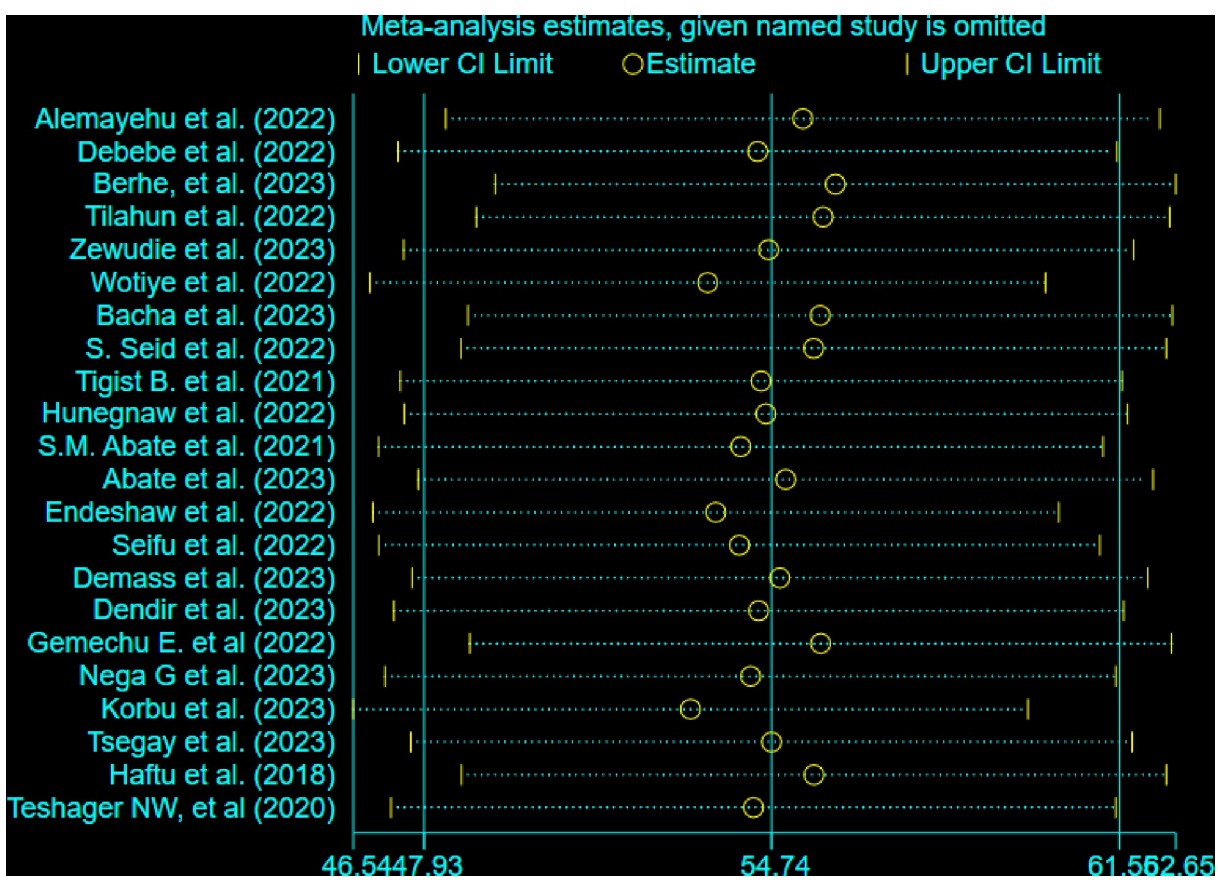

**Fig 8. Sensitivity analysis of mortality among mechanically ventilated ICU patients in Ethiopia, 2023.**

### Strength and limitation

To ensure that the findings were thorough and representative, this systematic review and meta-analysis attempted to encompass all available information, including published and gray literature, as well as cross-sectional and cohort study designs from Ethiopia. However, the included research covers only five of Ethiopia's more than nine regions. As a result, considering the limitations inherent in the original studies would be advantageous to a more accurate interpretation of the results.

### Conclusion

The pooled prevalence of mortality among mechanically ventilated ICU patients in Ethiopia was determined to be 54.74%, according to our review. This conclusion is significantly high compared to similar research in the US, China, and other countries. Sepsis, GCS<8 during admission to the ICU, medical cases, MODS, and the use of vasopressors were statistically associated with mortality. Clinicians should exercise caution while mechanically ventilating ICU-admitted patients with sepsis, who are unconscious with GCS<8, who have a medical case diagnosis, and who receive vasopressor medications. However, it is important to note that this meta-analysis could not demonstrate an accurate cause-and-effect association because the included studies only observe and record events without manipulating variables, and the available evidence is insufficient. Thus, more studies using prospective methods will be required to

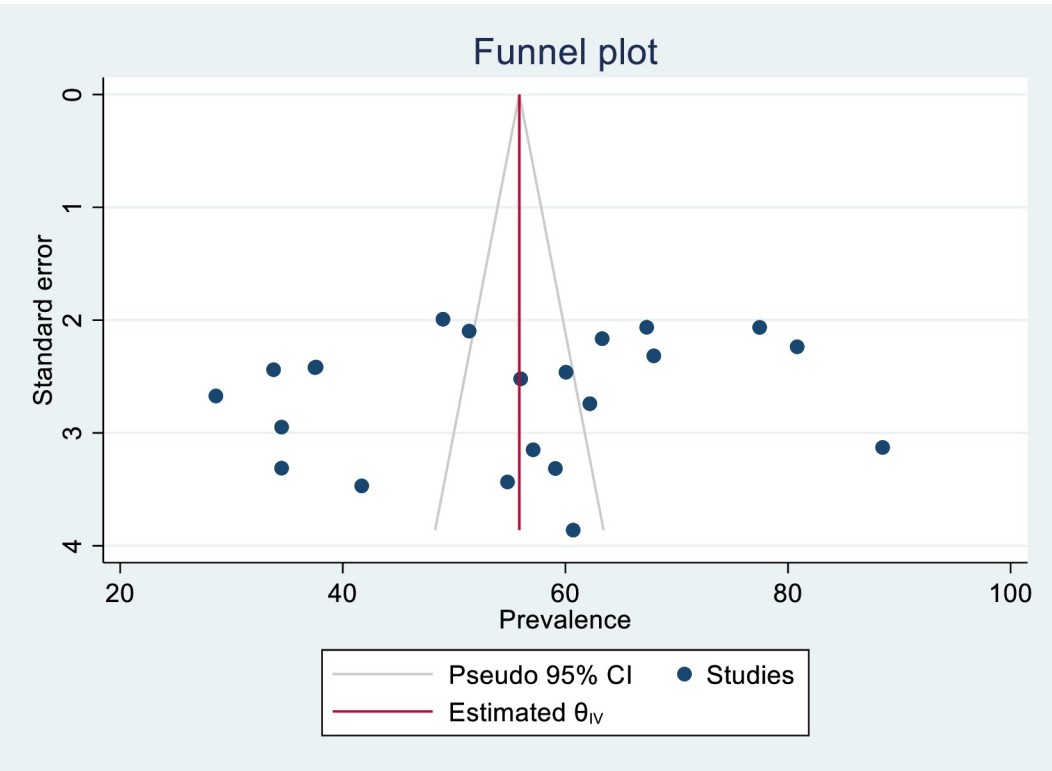

**Fig 9. Funnel plot of mortality among mechanically ventilated ICU patients in Ethiopia, 2023.**

**Table 2. Meta-regression analysis of factors affecting between study heterogeneity.**

| Hetroginity source | Coefficients | Standard error | p-valuse |
|---|---|---|---|
| Publication year | .9893661 | 3.08101 | 0.752 |
| Sample size | -.0053366 | .0274522 | 0.848 |
| JBI Quality | -2.507077 | 4.303919 | 0.567 |

**Table 3. Factors affecting mortality of mechanically ventilated ICU patients in Ethiopia, 2023.**

| Predictors | Number of studies | OR (95% CI) | $I^2$ (p value) |
|---|---|---|---|
| Sepsis | 2 [12, 24] | 6.85 (3.24, 14.46) | 0.0% (0.349) |
| GCS<8 | 4 [16, 17, 24, 36] | 6.58 (1.96, 22.11) | 93.4% (0.000) |
| ICU admission with medical cases | 4 [15–17, 36] | 4.12 (2.00, 8.48) | 85.1% (0.000) |
| MODS | 3 [15, 17, 24] | 2.70 (4.11, 12.62) | 46.6% (0.154) |
| Vassopressor treatment | 2 [12, 24] | 19.06 (9.34, 38.88) | 0.0% (0.0975) |

determine the factors that contribute to mortality in ICU-admitted patients on mechanical ventilation.

## Supporting information

**S1 Checklist. PRISMA 2020 checklist.**
(DOCX)

**S1 Table. Logic grid and search strategy for the systematic review of mortality of mechanically ventilated patients in intensive care units of Ethiopian hospitals, 2023.**
(DOCX)

**S1 Data set. The minimal data set used for the values behind the statistical measures reported and the values used to build graphs.**
(XLSX)

## Acknowledgments

Our grattitude goes to all individual at Debre Markos University, College of Health Sciences and School of Medicine, who assisted us in this review.

## Author Contributions

**Conceptualization:** Temesgen Ayenew, Mengistu Abebe Meselu.

**Data curation:** Mihretie Gedfew, Mamaru Getie Fetene, Belayneh Shetie Workneh, Animut Takele Telayneh, Afework Edmealem, Addisu Getie, Mengistu Abebe Meselu.

**Formal analysis:** Temesgen Ayenew, Mihretie Gedfew, Animut Takele Telayneh, Afework Edmealem, Addisu Getie, Mengistu Abebe Meselu.

**Investigation:** Mamaru Getie Fetene, Addisu Getie.

**Methodology:** Temesgen Ayenew, Mihretie Gedfew, Belayneh Shetie Workneh, Animut Takele Telayneh, Addisu Getie, Mengistu Abebe Meselu.

**Software:** Temesgen Ayenew, Belayneh Shetie Workneh.

**Validation:** Temesgen Ayenew.

**Visualization:** Mihretie Gedfew, Afework Edmealem, Bekele Getenet Tiruneh, Guadie Tewabe Yinges.

**Writing – original draft:** Temesgen Ayenew, Mengistu Abebe Meselu.

**Writing – review & editing:** Bekele Getenet Tiruneh, Guadie Tewabe Yinges.

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
