## [Decision Letter · Decision Letter 0]

17 Apr 2024

PONE-D-24-00386Prevalence of Mortality of Mechanically Ventilated Patients and Associative Factors in Intensive Care Units of Ethiopian Hospitals: A Systematic Review and Metanalysis.PLOS ONE

Dear Dr. Ayenew,

Thank you for submitting your manuscript to PLOS ONE. After careful consideration, we feel that it has merit but does not fully meet PLOS ONE’s publication criteria as it currently stands. Therefore, we invite you to submit a revised version of the manuscript that addresses the points raised during the review process.

We look forward to receiving your revised manuscript.

Kind regards,

Kahsu Gebrekidan

Academic Editor

PLOS ONE

Journal Requirements:

Reviewers' comments:

Reviewer's Responses to Questions

**Comments to the Author**

1. Is the manuscript technically sound, and do the data support the conclusions?

Reviewer #1: Yes

Reviewer #2: Partly

2. Has the statistical analysis been performed appropriately and rigorously? 

Reviewer #1: Yes

Reviewer #2: I Don't Know

3. Have the authors made all data underlying the findings in their manuscript fully available?

Reviewer #1: Yes

Reviewer #2: Yes

4. Is the manuscript presented in an intelligible fashion and written in standard English?

Reviewer #1: Yes

Reviewer #2: Yes

5. Review Comments to the Author

Reviewer #1: The article is simple and straightforward and the topic is interesting to read. The abstract summarizes the paper well and the intro clarifies the objective and rationale behind the study. The method is adequately described. The strategy behind the review and meta analysis was clearly described. Risk of bias was assessed. Variables, outcomes and associative factors were clearly shown and explained. Data were consistent with findings and limitations were transparently discussed.

I have one question, I was curious as to why the number of included studies became 22 for the final systematic review and meta-analysis? Based on what did you exclude the last 4 studies from the 26 studies? It would be great if you could clarify that.

Also, I would suggest adding legends to the figures.

Thank you for you efforts in writing this paper.

Reviewer #2: The text discusses a systematic review that aims to answer a frequently asked question, but it is limited to a specific country that the authors belong to. Although several publications have addressed this question, the review was conducted following the PRISMA guidelines for systematic reviews and meta-analyses. However, certain areas require clarification, as the article may not be entirely clear.

Below are some areas or points that need clarification, revision or highlighting.

The abstract section:

Establishing cause-and-effect relationships can be challenging when conducting a systematic review using observational studies. These studies merely observe and record events without manipulating variables. Therefore, it is crucial to explicitly state this limitation in the abstract's conclusion, refraining from making claims that the available evidence cannot substantiate.

Methods section

The study's inclusion criteria were carefully designed to be specific and broad enough to cover a wide range of patients. Although the mortality rates between adult and pediatric ICU patients were significantly different, age differences were not considered a distinguishing factor to ensure a comprehensive review that encompasses all relevant studies.

The study's secondary outcome was not specified or mentioned in the methods sections, making it unclear which factors the systematic review will focus on.

Result section :

For subgroup analysis, it may be helpful to consider the study type (cohort or cross-sectional) and age differences (adults and pediatrics).

Why did they choose this publication and sample size to check the cause of heterogeneity of the article?

The weight of each study was around 4. As I am not a statistician, I wonder how they get similar weights for different studies with varying populations and sample sizes.

The secondary outcome in the Results section should be rewritten. I’m not sure if the OR is adjusted or the raw OR.

Discussion section :

Most of the area is covered. However, the final conclusion needs to be rewritten. The first part concerning prevalence is acceptable, but the second part needs to be reworded.

6. PLOS authors have the option to publish the peer review history of their article (what does this mean?). If published, this will include your full peer review and any attached files.

Reviewer #1: No

Reviewer #2: No

---

## [Author Response · Author response to Decision Letter 0]

22 Apr 2024

A point-by-point response to review comments to the author

Dear Editor and reviewers, we are very grateful for your constructive comments to our submitted manuscript. Your comments are of great value. We have addressed them point by point in the revised manuscript. The revisions are indicated with tracked changes in the revised manuscript. Here below is a table showing point by point responses to your comments.

Editor’s comments

• Thank you for your feedback. It is corrected as recommended.

Please include captions for your Supporting Information files at the end of your manuscript, and update any in-text citations to match accordingly.

• Thank you for your feedback. It is corrected as recommended in the revised version

Reviewer #1

I was curious as to why the number of included studies became 22 for the final systematic review and meta-analysis? Based on what did you exclude the last 4 studies from the 26 studies? It would be great if you could clarify that.

• Thank you so much, we have explained it in the revised manuscript on page 6

I would suggest adding legends to the figures

• Thank you, figure legends are incorporated throughout the whole manuscript

Reviewer #2

Abstract

Establishing cause-and-effect relationships can be challenging when conducting a systematic review using observational studies. These studies merely observe and record events without manipulating variables. Therefore, it is crucial to explicitly state this limitation in the abstract's conclusion, refraining from making claims that the available evidence cannot substantiate.

• Thank you so much, we really appreciate your comments. It is addressed in the revised manuscript at the conclusion of the abstract section and at the conclusion section of the manuscript on pages 3 & 13

Methods

The study's inclusion criteria were carefully designed to be specific and broad enough to cover a wide range of patients. Although the mortality rates between adult and pediatric ICU patients were significantly different, age differences were not considered a distinguishing factor to ensure a comprehensive review that encompasses all relevant studies. 

• Thank you in advance, actually we didn’t restrict our inclusion criteria using age category. So, we have revised the inclusion criteria to include a statement “both pediatric and adult ICU-admitted patients” to make it clearer for readers that we had considered age categories on page 5

The study's secondary outcome was not specified or mentioned in the methods sections, making it unclear which factors the systematic review will focus on. 

• Dear reviewer, thank you so much. We have addressed it in the revised manuscript on page 5.

Results 

For subgroup analysis, it may be helpful to consider the study type (prospective cohort, retrospective cohort and cross-sectional) and age differences (adults and pediatrics). 

• Thank you so much, sub-group analysis was done using these variables in the revised manuscript on page 8

Why did they choose this publication and sample size to check the cause of heterogeneity of the article? 

• Dear reviewer, thank you so much. we use sample size, JBI quality and publication year to run meta-regression analysis. Meta-regression analysis command uses only integer variables and do not allow string variables like region and study design on pages 9-10

The weight of each study was around 4. As I am not a statistician, I wonder how they get similar weights for different studies with varying populations and sample sizes. 

• Dear reviewer, thank you for your concerns. Yes, of course, the weight of each study is almost close to 4. This might be because we used a random effects model. Under the random-effects model the goal is not to estimate one true effect, but to estimate the mean of a distribution of effects. Since each study provides information about a different effect size, we want to be sure that all these effect sizes are represented in the summary estimate. This means that we cannot discount a small study by giving it a very small weight. The estimate provided by that study may be imprecise, but it is information about an effect that no other study has estimated. By the same logic we cannot give too much weight to a very large study. Our goal is to estimate the mean effect in a range of studies and we do not want that overall estimate to be overly influenced by any one of them.

The secondary outcome in the Results section should be rewritten. I’m not sure if the OR is adjusted or the raw OR. 

• Dear reviewer, thank you for your suggestion. The odds ratio is the pooled odds ratio (adjusted) after meta-analysis of significant variables from individual studies.

Discussion

Most of the area is covered. However, the final conclusion needs to be rewritten. The first part concerning prevalence is acceptable, but the second part needs to be reworded. 

• Dear reviewer, thank you for your suggestion, we have made revisions in the conclusion on page 13

Finally, thank you so much again!

---

## [Decision Letter · Decision Letter 1]

7 May 2024

PONE-D-24-00386R1Prevalence of mortality of mechanically ventilated patients and associated factors in intensive care units of Ethiopian hospitals: systematic review and meta-analysisPLOS ONE

Dear Dr. Ayenew,

Thank you for submitting your manuscript to PLOS ONE. After careful consideration, we feel that it has merit but does not fully meet PLOS ONE’s publication criteria as it currently stands. Therefore, we invite you to submit a revised version of the manuscript that addresses the points raised during the review process.

We look forward to receiving your revised manuscript.

Kind regards,

Kahsu Gebrekidan

Academic Editor

PLOS ONE

Journal Requirements:

Reviewers' comments:

Reviewer's Responses to Questions

**Comments to the Author**

1. If the authors have adequately addressed your comments raised in a previous round of review and you feel that this manuscript is now acceptable for publication, you may indicate that here to bypass the “Comments to the Author” section, enter your conflict of interest statement in the “Confidential to Editor” section, and submit your "Accept" recommendation.

Reviewer #2: All comments have been addressed

Reviewer #3: (No Response)

2. Is the manuscript technically sound, and do the data support the conclusions?

Reviewer #2: Yes

Reviewer #3: Yes

3. Has the statistical analysis been performed appropriately and rigorously? 

Reviewer #2: I Don't Know

Reviewer #3: Yes

4. Have the authors made all data underlying the findings in their manuscript fully available?

Reviewer #2: Yes

Reviewer #3: Yes

5. Is the manuscript presented in an intelligible fashion and written in standard English?

Reviewer #2: Yes

Reviewer #3: Yes

6. Review Comments to the Author

**Reviewer #2:** (No Response)

**Reviewer #3**: Suggest to rephrase the title to "Prevalence of mortality among mechanically ventilated patients in the intensive care units of Ethiopian hospitals and the associated factors: A systematic review and meta-analysis"

Line 85&86: "These include the presence of comorbidities, length of stay on MV, day

and time of ICU admission, Glasgow coma scale (GCS) during admission" please specify the length of stay e.g more than seven days, and the GCS score of 11 and above (for example).

line 185: please correct the spelling of heterogeneity

Please check the journal format for abbreviating the word Figure, is it "Fig 2" or "Fig. 2"

Line 247: please write in full " is not"

7. PLOS authors have the option to publish the peer review history of their article (what does this mean?). If published, this will include your full peer review and any attached files.

Reviewer #2: No

Reviewer #3: No

---

## [Author Response · Author response to Decision Letter 1]

8 May 2024

Response to Reviewers

Dear editor and reviewers, thank you for your constructive comments for the second time. We value your comments and we have addressed them point by point in the revised manuscript. The revisions are indicated with tracked changes in the revised manuscript. Here below is a point-by-point response to the reviewers and the editor.

Journal Requirements

• Dear editor, thank you for your comments regarding the referencing style. We have made revisions. All of the references are changed to Vancouver style after seeing the PLOS Submission Guidelines references section on pages 14 – 18. No retracted articles were cited.

Review Comments to the Author

Reviewer #2: (No Response)

Reviewer #3: Suggest to rephrase the title to "Prevalence of mortality among mechanically ventilated patients in the intensive care units of Ethiopian hospitals and the associated factors: A systematic review and meta-analysis"

• Dear reviewer, thank you for your suggestion. The title is rephrased accordingly on pages 1 & 2

Line 85&86: "These include the presence of comorbidities, length of stay on MV, day

and time of ICU admission, Glasgow coma scale (GCS) during admission" please specify the length of stay e.g more than seven days, and the GCS score of 11 and above (for example).

• Dear reviewer we value your comments, revisions are made to make these specific on page 4

line 185: please correct the spelling of heterogeneity

• Dear reviewer, thank you for your comments, the spelling is corrected on page 8

Please check the journal format for abbreviating the word Figure, is it "Fig 2" or "Fig. 2"

• Dear reviewer, thank you for your comments, but according to manuscript body formatting guidelines of the journal figures are labeled as “Fig” not “Fig.” (https://journals.plos.org/plosone/s/file?id=wjVg/PLOSOne_formatting_sample_main_body.pdf) 

Line 247: please write in full " is not"

• Dear reviewer, thank you for your comments, the word is written in its full form as “is not” on page 11

Finally, thank you in advance for your prompt and constructive feedbacks.

Sincerely,

Temesgen Ayenew (corresponding author)

---

## [Decision Letter · Decision Letter 2]

4 Jun 2024

PONE-D-24-00386R2Prevalence of mortality among mechanically ventilated patients in the intensive care units of Ethiopian hospitals and the associated factors: A systematic review and meta-analysisPLOS ONE

Dear Dr. Ayenew,

Thank you for submitting your manuscript to PLOS ONE. After careful consideration, we feel that it has merit but does not fully meet PLOS ONE’s publication criteria as it currently stands. Therefore, we invite you to submit a revised version of the manuscript that addresses the points raised during the review process.

Please submit your revised manuscript by Jul 19 2024 11:59PM If you will need more time than this to complete your revisions, please reply to this message or contact the journal office at plosone@plos.org. Please include the following items when submitting your revised manuscript:A rebuttal letter that responds to each point raised by the academic editor and reviewer(s). You should upload this letter as a separate file labeled 'Response to Reviewers'.A marked-up copy of your manuscript that highlights changes made to the original version. You should upload this as a separate file labeled 'Revised Manuscript with Track Changes'.An unmarked version of your revised paper without tracked changes. You should upload this as a separate file labeled 'Manuscript'.If applicable, we recommend that you deposit your laboratory protocols in protocols.io to enhance the reproducibility of your results. Protocols.io assigns your protocol its own identifier (DOI) so that it can be cited independently in the future. For instructions see: https://journals.plos.org/plosone/s/submission-guidelines#loc-laboratory-protocols. Additionally, PLOS ONE offers an option for publishing peer-reviewed Lab Protocol articles, which describe protocols hosted on protocols.io. Read more information on sharing protocols at https://plos.org/protocols?utm_medium=editorial-email&utm_source=authorletters&utm_campaign=protocols.

We look forward to receiving your revised manuscript.

Kind regards,

Kahsu Gebrekidan

Academic Editor

PLOS ONE

Journal Requirements:

Reviewers' comments:

Reviewer's Responses to Questions

**Comments to the Author**

1. If the authors have adequately addressed your comments raised in a previous round of review and you feel that this manuscript is now acceptable for publication, you may indicate that here to bypass the “Comments to the Author” section, enter your conflict of interest statement in the “Confidential to Editor” section, and submit your "Accept" recommendation.

Reviewer #3: (No Response)

2. Is the manuscript technically sound, and do the data support the conclusions?

Reviewer #3: Yes

3. Has the statistical analysis been performed appropriately and rigorously? 

Reviewer #3: Yes

4. Have the authors made all data underlying the findings in their manuscript fully available?

Reviewer #3: Yes

5. Is the manuscript presented in an intelligible fashion and written in standard English?

Reviewer #3: Yes

6. Review Comments to the Author

Reviewer #3: Dear Authors,

1) Please check spelling error throughout the article. I found a few, with in (should be within), and line 186 (spelling for heterogeneity), line 194 (spelling for yields).

2) Please check the grammar: I found error in line 272 and 273. based on the references (47& 48), there were two separate studies, therefore should use support (without s).

7. PLOS authors have the option to publish the peer review history of their article (what does this mean?). If published, this will include your full peer review and any attached files.

Reviewer #3: No

---

## [Author Response · Author response to Decision Letter 2]

6 Jun 2024

Response to Reviews

Dear Reviewer, many thanks to your constructive feedbacks. We have extensively seen the whole manuscript for spelling and grammar errors. Here below is a point-by-point response to the reviewer’s comments.

Reviewer #3

1. Please check spelling error throughout the article. I found a few, with in (should be within), and line 186 (spelling for heterogeneity), line 194 (spelling for yields).

Dear reviewer, thank you for your comments, we have made spelling corrections to these words as well as other spelling errors throughout the manuscript. The corrections are indicated with tracked changes.

2. Please check the grammar: I found error in line 272 and 273. based on the references (47& 48), there were two separate studies, therefore should use support (without s).

Dear reviewer, thank you for your comments, we have made grammar checks and made corrections to these sentence and others throughout the article. The corrections are indicated with tracked changes

We really appreciate your feedback

Temesgen Ayenew, 

corresponding author

---

## [Editor Report · Decision Letter 3]

15 Jun 2024

Prevalence of mortality among mechanically ventilated patients in the intensive care units of Ethiopian hospitals and the associated factors: A systematic review and meta-analysis

PONE-D-24-00386R3

Dear Mr. Temesgen,

We’re pleased to inform you that your manuscript has been judged scientifically suitable for publication and will be formally accepted for publication once it meets all outstanding technical requirements.

Kind regards,

Kahsu Gebrekidan

Academic Editor

PLOS ONE
---

## [Editor Report · Acceptance letter]

18 Jun 2024

PONE-D-24-00386R3 

PLOS ONE

Dear Dr. Ayenew, 

I'm pleased to inform you that your manuscript has been deemed suitable for publication in PLOS ONE. Congratulations! Your manuscript is now being handed over to our production team.

Kind regards, 

on behalf of

Dr. Kahsu Gebrekidan 

Academic Editor

PLOS ONE